# Bioengineered Carboxymethylcellulose–Peptide Hybrid Nanozyme Cascade for Targeted Intracellular Biocatalytic–Magnetothermal Therapy of Brain Cancer Cells

**DOI:** 10.3390/pharmaceutics14102223

**Published:** 2022-10-18

**Authors:** Alexandra A. P. Mansur, Sandhra M. Carvalho, Luiz Carlos A. Oliveira, Elaine Maria Souza-Fagundes, Zelia I. P. Lobato, Maria F. Leite, Herman S. Mansur

**Affiliations:** 1Center of Nanoscience, Nanotechnology, and Innovation—CeNano2I, Department of Metallurgical and Materials Engineering, Engineering School, Federal University of Minas Gerais (UFMG), Av. Antônio Carlos, 6627, Belo Horizonte 31270-901, MG, Brazil; 2Departament of Chemistry, Federal University of Minas Gerais (UFMG), Belo Horizonte 31270-901, MG, Brazil; 3Department of Physiology and Biophysics, Institute of Biological Sciences-ICB, Federal University of Minas Gerais (UFMG), Belo Horizonte 31270-901, MG, Brazil; 4Veterinary School, Federal University of Minas Gerais (UFMG), Belo Horizonte 31270-901, MG, Brazil

**Keywords:** nanoparticles, tumor targeting, cancer, theranostics, nanotheranostics, nanohybrids, polymer–peptide bioconjugates, nanozyme biocatalytic cascade

## Abstract

Glioblastoma remains the most lethal form of brain cancer, where hybrid nanomaterials biofunctionalized with polysaccharide peptides offer disruptive strategies relying on passive/active targeting and multimodal therapy for killing cancer cells. Thus, in this research, we report for the first time the rational design and synthesis of novel hybrid colloidal nanostructures composed of gold nanoparticles stabilized by trisodium citrate (AuNP@TSC) as the oxidase-like nanozyme, coupled with cobalt-doped superparamagnetic iron oxide nanoparticles stabilized by carboxymethylcellulose ligands (Co-MION@CMC) as the peroxidase-like nanozyme. They formed inorganic–inorganic dual-nanozyme systems functionalized by a carboxymethylcellulose biopolymer organic shell, which can trigger a biocatalytic cascade reaction in the cancer tumor microenvironment for the combination of magnetothermal–chemodynamic therapy. These nanoassemblies were produced through a *green* aqueous process under mild conditions and chemically biofunctionalized with integrin-targeting peptide (iRDG), creating bioengineered nanocarriers. The results demonstrated that the oxidase-like nanozyme (AuNP) was produced with a crystalline face-centered cubic nanostructure, spherical morphology (diameter = 16 ± 3 nm), zeta potential (ZP) of −50 ± 5 mV, and hydrodynamic diameter (D_H_) of 15 ± 1 nm. The peroxide-like nanostructure (POD, Co-MION@CMC) contained an inorganic crystalline core of magnetite and had a uniform spherical shape (2R = 7 ± 1 nm) which, summed to the contribution of the CMC shell, rendered a hydrodynamic diameter of 45 ± 4 nm and a negative surface charge (ZP = −41 ± 5 mV). Upon coupling both nanozymes, water-dispersible colloidal supramolecular vesicle-like organic–inorganic nanostructures were produced (AuNP//Co-MION@CMC, ZP = −45 ± 4 mV and D_H_ = 28 ± 3 nm). They confirmed dual-nanozyme cascade biocatalytic activity targeted by polymer–peptide conjugates (AuNP//Co-MION@CMC_iRGD, ZP = −29 ± 3 mV and D_H_ = 60 ± 4 nm) to kill brain cancer cells (i.e., bioenergy “*starvation*” by glucose deprivation and oxidative stress through reactive oxygen species generation), which was boosted by the magneto-hyperthermotherapy effect when submitted to the alternating magnetic field (i.e., induced local thermal stress by “*nanoheaters*”). This groundwork offers a wide avenue of opportunities to develop innovative theranostic nanoplatforms with multiple integrated functionalities for fighting cancer and reducing the harsh side effects of conventional chemotherapy.

## 1. Introduction

Despite extraordinary progress in cancer diagnosis and therapy in recent years, glioblastoma (GBM) remains the deadliest form of brain cancer, causing over 241,000 deaths worldwide in 2018 [1,2]. Regrettably, the currently available treatments mostly rely on conventional therapies, involving surgical resection, chemotherapy, radiotherapy, and sometimes combining two or more approaches. Thus, nanotechnology associated with medicine—termed nanomedicine—can drastically alter this scenario by offering innovative advanced multifunctional nanomaterials for fighting cancer [1,2].

In this sense, chemodynamic therapy (CDT) based on Fenton or Fenton-like reactions is an emerging treatment that can effectively fight cancer and reduce side effects on normal cells and tissues. CDT is based on the transformation of endogenous H_2_O_2_ through Fenton or Fenton-like reactions into highly harmful hydroxyl radicals (•OH), known as the most oxidizing reactive oxygen species (ROS), which can induce massive cell death. Especially as the content of hydrogen peroxide (H_2_O_2_) in tumor tissues is far higher than that in normal tissues, CDT relies on this feature to improve the selectivity, reducing the side effects on normal tissues. However, the concentration of H_2_O_2_ in the tumor microenvironment (TME) is insufficient to continuously produce •OH. Therefore, CDT can be significantly improved via accelerating the Fenton-based reactions by increasing the local concentration of H_2_O_2_ in cancer cells. In addition to the above strategies, multimodal cancer therapy is another direct way to augment the therapeutic effect of CDT. For example, CDT can be combined with photothermal therapy (PTT), photodynamic therapy (PDT), hyperthermia therapy (HT), and/or sonodynamic therapy (SDT). Nevertheless, as PTT and PDT rely on light as external energy to enhance the therapeutic effect, their application is limited because of their low maximum penetration depth in the body (about 10 mm). Thus, this restriction stimulates the use of alternative treatments based on raising tumor temperature that present higher penetration ability, such as microwave thermal therapy and magnetothermal therapy. This kind of therapy takes the benefits of the low temperature resistance of cancer cells compared to normal human cells and tissues and, when combined with CDT, can boost the catalytic activity of the Fenton agents [3,4,5].

Several nanostructures have been designed and developed to amalgamate these multimodal functionalities, including hybrid nanomaterials (nanohybrids). Nanohybrids are a novel class of nanomaterials, mainly composed of two or more materials combining organic and inorganic components, and have emerged as promising candidates for revolutionizing oncological nanomedicine applications. They combine the original *soft matter* and *hard matter* functions of each component, endowing multifunctional stimulus-responsive features to hybrid nanoarchitectures [6,7].

The *soft matter* portion of the hybrid nanostructures, often applied as the shell layer, is typically composed of organic components such as polymer macromolecules, biomolecules, and bioconjugates, which play a crucial role in the chemical stability of the nanosystems. They also ascribe the biological functionalization for affinity recognition of cancerous cells, diseased tissues, and organs [8,9]. Polymer-based nanoassemblies—especially biopolymers—have been preferred as *soft matter* for building hybrid supramolecular nanostructures that possess the ability to undergo dynamic/reversible changes in conformation, shape, and charge distribution, along with excellent functions in response to chemical stimuli (i.e., pH, concentration, ionic strength, charge potential) for oncological nanomedicine applications [7,10]. Hence, biological and pharmaceutical carbohydrate-based systems—including nanoparticles, hydrogels, 3D scaffolds, and organic shell layers—have increasingly received attention for several advanced applications (e.g., drug delivery, bioimaging, tissue engineering, nanotheranostics, etc.) because they are the most abundant, biocompatible, and essential biomacromolecules in nature [2,6,7,8,9,10,11,12,13]. Consequently, polysaccharides (e.g., hyaluronic acid, chitosan, cellulose, and their derivatives) have been the most common choice of nature-sourced biocompatible polymers. Compared with other biopolymers such as proteins, polysaccharide-based applications are usually biocompatible and non-toxic; they are typically biodegradable, and the released monosaccharide is free of toxicity [14]. Moreover, they have several derivable groups on their molecular chains—especially carboxymethylcellulose (CMC), which can be easily adapted and modified with other chemical agents. CMC presents exceptional biodegradability, non-toxicity, suitable reactivity for simple chemical modification associated with global availability, affordable cost, and approval by the United States Food and Drug Administration (FDA) for biotechnology applications and biomedicine [7,8,9]. Additionally, CMC polymers encompass passive targeting characteristics via enhanced permeability and retention (EPR) effects regarding the benefits of nanomedicine. They possess reactive chemical groups (e.g., hydroxyls and carboxyls), which permit their tunable functionalization with affinity biomolecules (e.g., amino acids, peptides, proteins, antibodies) for developing active targeting through surface-conjugated biomolecules. Principally, RGD-based peptides containing the triple amino acid sequence Arg-Gly-Asp (e.g., iRGD, a novel cyclic peptide composed of nine amino acids including an RGD motif) have been widely used for producing polymer-based drug delivery platforms. They present precise targeting toward cell membrane receptors (i.e., α_v_β_3_ and α_v_β_5_ integrins) that are overexpressed by several cancerous tumors, including the GBM form [15,16,17,18]. This approach permits optimizing the efficiency of cancer therapies by lowering drug concentrations and surpassing one of the main challenges in nano-oncology research associated with undesired side effects, including the non-specific attack of the chemotherapeutic molecules on normal cells when using conventional treatments [8,9,19].

Considering the second portion of the hybrid nanosystems, the *hard matter* is generally composed of inorganic nanomaterials, including colloidal semiconductor quantum dots (QDs) [2,7,9], metallic nanoparticles (e.g., Au, Ag, Pt) [6,20], superparamagnetic iron oxide nanoparticles (SPIONs or MIONs), and other nanosized compounds [6,19]. In this realm, recently, nanozymes—a variety of natural-enzyme-mimicking inorganic nanomaterials—have emerged as an innovative area of research that can play a pivotal role in oncological nanomedicine for diagnosis and therapy [21,22,23]. Compared with natural enzymes, nanozymes have attracted widespread interest due to their superior thermal and chemical stability, high catalytic efficiency, biosafety, relatively low cost, and easy preparation. Essentially, in nano-oncology therapy, nanozymes can kill cancer cells in two ways: The first is direct killing by increasing reactive oxygen species (ROS) through the oxidase and peroxidase activities of nanozymes. The second is indirect killing, related to the catalase or superoxide dismutase activity of nanozymes, by relieving the hypoxia of the tumor microenvironment [22,24]. Three major groups of materials are associated with nanozymes’ mimicking behavior: metal-compound-based (e.g., Fe_3_O_4_, Co_3_O_4_, CeO_2_), noble-metal-based (e.g., Au, Ag, Pt, Pd), and carbon-based (e.g., graphene oxide, carbon dots, and carbon nanotubes) [25]. Among these alternatives, iron-oxide-based nanomaterials are the most intensively researched nanozymes associated with the peroxidase-like (POD) activity of ferromagnetic nanoparticles. Interestingly, superparamagnetic magnetite, with or without doping, is also one of the most attractive materials for MHT due to the combination of high saturation magnetization and magnetic anisotropy [26].

Thus, by designing and constructing nanohybrids, one can amalgamate the advantages of polymer–peptide (i.e., *soft matter*) conjugates for targeting cancer cells with nanozymes (i.e., *hard matter*) to promote ROS generation and temperature increase by an alternated magnetic field, inducing cancer cell death pathways [19,22]. In this view, by adopting a similar approach by coupling two or more nanozymes in a sequence of catalytic cascade reactions, one can offer unlimited opportunities to create hybrid nanomedicine platforms with multiple stimulus-responsive structures for fighting against cancer [22,23,24,25,27].

Remarkably, although it is a fascinating realm of research, the area of multi-nanozyme catalytic cascades for oncological nanomedicine and nanotheranostics has scarcely been investigated. Hence, as no report was found in the consulted literature, here we hypothesize for the first time that nanohybrids composed of two inorganic nanozymes (i.e., gold nanoparticles and Co-doped magnetite nanohybrids) could be designed, synthesized, and functionalized by CMC polysaccharide using an eco-friendly aqueous colloidal route, which would be engineered to perform biocatalytic cascade activity combined with magnetic hyperthermia features. Moreover, these nanoconjugates could be biofunctionalized with the iRGD peptide to render colloidal supramolecular nanoassemblies, effectively targeting and killing live brain cancer cells in vitro through an enhanced chemocatalytic–magnetothermotherapy process (i.e., oxidase/peroxidase-like endogenous biocatalytic activity for accelerated Fenton-like reactions and magnetic hyperthermia).

## 2. Experimental Procedure

All materials, cell cultures, and more conventional experimental methods and protocols are described in the Appendix A to avoid redundancy.

The synthesis of the cobalt-doped iron oxide was based on the co-precipitation method using carboxymethylcellulose (CMC) as a stabilizing agent in alkaline conditions, as previously described by our group [26]. The cobalt doping was performed by replacing 10 mol% Fe^2+^ from stoichiometric Fe_3_O_4_ with Co^2+^ ions and forming CMC-functionalized nanoconjugates, referred to as Co-MION@CMC (Fe_3_O_4_ concentration of 2.4 mg/mL).

The Au nanoparticles (termed AuNPs) were obtained in an aqueous medium based on in situ reduction and stabilization by trisodium citrate (TSC) at a concentration of 40 μg/mL in TSC solution (1.2 mM) [28]. This suspension was centrifuged and resuspended to achieve the 100 μg/mL concentration, which was used to prepare the dual nanozymes (binanozymes).

The hybrid nanoassemblies composed of two integrated inorganic nanozymes (AuNP//Co-MION@CMC, referred to as dual nanozymes or binanozymes) were obtained as follows: First, 370 µL of Co-MION@CMC suspension (2.4 mg/mL) was dispersed into 600 µL of TSC solution (1.2 mM). Then, 2 mL of AuNP suspension (100 µg/mL) was added dropwise to an iron oxide suspension under an ultrasonic bath and sonicated for 2 h at 25 ± 2 °C. The mass ratio of AuNP:Co-MION was about 1:4.3. This suspension was stored at 6 ± 2 ˚C until further use.

Biofunctionalization of the dual-nanozyme structure with the iRGD peptide was performed using 1-ethyl-3-[3-dimethylaminopropyl]carbodiimide hydrochloride (EDC) and N-hydroxysulfosuccinimide (sulfo-NHS). The peptide-functionalized dual-nanozyme nanoassemblies were named AuNP//Co-MION@CMC_iRGD or Bi-nano_iRGD. For the preparation of the Bi-nano_iRGD nanoconjugate, 200 μL of EDC solution (0.64%, *w*/*v*) and 200 μL of sulfo-NHS (1.6%, *w*/*v*) were added to 1.5 mL of Co-MION@CMC//AuNP suspension (pH adjusted 5.4 ± 0.2, HCl 0.1 M) and magnetically stirred for 15 min at 6 ± 2 °C. Then, 1.0 mL of iRGD solution (0.03 %, *w*/*v* with pH adjusted to 7.1 ± 0.2, HCl 0.1 M) was added to the flask, and the system was incubated at 6 ± 2 °C for 2 h in the dark under magnetic stirring. The molar ratio of the CMC-COO^−^ group:iRGD added was 10:1.

The nanoparticles/nanostructures were extensively characterized using several techniques to evaluate their spectroscopic, structural, and morphological features, as well as their physicochemical properties, including Fourier-transform infrared spectroscopy (FTIR), high-resolution X-ray photoelectron spectroscopy (HR-XPS), ultraviolet–visible spectroscopy (UV–Vis), X-ray fluorescence spectrometry (WD-XRF), dynamic light scattering (DLS), zeta potential (ZP), and transmission electron microscopy (TEM) coupled with energy-dispersive X-ray spectroscopy (EDS) (see the Appendix A for details of the characterization procedures and equipment).

The in vitro acellular catalytic functionality of the inorganic nanozymes (Co-MION@CMC and AuNP) was investigated based on the oxidation of 3,3′,5,5′-tetramethylbenzidine hydrochloride (TMB → TMB_ox_), whose bluish color was detected by UV–Vis spectroscopy at λ  =  645−655 nm. The oxidation reaction to evaluate the oxidase-like behavior of the gold nanoparticles was mediated by the horseradish peroxidase (HRP) enzyme.

Biological tests were performed using U87 glioblastoma cancer cells and HEK 293T healthy cells (human embryonic kidney cells). Human brain likely glioblastoma cells (U87, American Type Culture Collection—ATCC^®^ HTB-14™) were selected because glioblastoma multiforme is one of the deadliest types of brain cancer, against which our innovative theranostic nanoplatform was investigated in this study. As a reference, human embryonic kidney cells (HEK 293T, ATCC^®^ CRL 1573) were selected to evaluate the prospective cytotoxic effects of the nanostructures in normal cells. This cell line is among the widely used standards for normal human cells in various biological experiments. It is justified because kidney toxicity is critical in clinical practice, as the kidneys play an important role in the body’s homeostasis and decide between renal clearance or accumulation of substances [29].

The nanoformulations were designed based on previously published papers by our group [20,26], considering the concentrations of AuNPs and Co-MION (10 mol% Fe^2+^ replaced by Co^2+^) associated with cell viability/toxicity in vitro, as depicted in Appendix A.

The intracellular level of “reactive oxygen species” (ROS) was estimated using 2′,7′-dichlorodihydrofluorescein diacetate (DCF-DA), as reported by our group [24]. After incubation with the nanosystems, the generation of fluorescent 2′,7′-dichlorofluorescein (DCF) species in both cell lines was measured using the emission intensity of DCF.

To evaluate the cellular in vitro nanocatalytic therapeutic efficiency, the cytotoxicity of the nanosystems was tested using the MTT (3-(4,5-dimethylthiazol-2yl-) 2,5-diphenyl tetrazolium bromide) protocol, as previously described by our group [8,9,24].

Lipid peroxidation was analyzed using the thiobarbituric acid method (TBA test) to determine malondialdehyde (MDA) via UV–Vis spectroscopy [30] after 24 h of contact with the nanostructures.

Regarding the magnetic and thermomagnetic behaviors of the cobalt-doped nanoconjugates, magnetic hysteresis curves were derived and magnetic hyperthermia tests under an alternating magnetic field (AMF) were performed in the nanocolloidal suspensions. The MTT bioassay measured the effects of magnetothermal therapy to quantify cell viability responses after incubating U87 cells for 3 h with the dual-nanozyme systems, followed by applying an AMF for 60 min [26].

## 3. Results and Discussion

*Rationale:* Design and synthesis of colloidal targeted biofunctionalized hybrid nanoassemblies

In this study, the primary aim was to combine a set of unique physicochemical, biochemical, and therapeutic properties offered by the amalgamation of polysaccharide peptides as the cellular targeting component (i.e., *soft matter*), along with inorganic nanozymes as the biocatalytic–magnetothermotherapeutic constituents (i.e., *hard matter*), integrated into an aqueous colloidal supramolecular nanostructure for multimodal cancer therapy. For clarity, these hybrid nanostructures were divided into two main subunits based on their nature and functions, as follows: (i) cobalt-doped iron oxide nanoparticles as the peroxidase-like nanozyme (POD), functionalized by carboxymethylcellulose biopolymer (Co-MION@CMC) for chemodynamic therapy (CDT) based on Fenton-like reactions, and magneto-hyperthermia therapy (MHT) related to their superparamagnetic properties; (ii) gold nanoparticles stabilized by trisodium citrate (TSC) as the glucose oxidase-like (GOD) nanozyme (AuNP) for producing H_2_O_2_ to accelerate the Fenton reaction. The complete process of designing and building these hybrid colloidal nanostructures is schematically depicted in Figure 1.

Firstly, carboxymethylcellulose polysaccharide (CMC) was used as the macromolecular surface-capping ligand for the nucleation and growth of the nanozyme Co-MION@CMC—which was required for controlling the formation of homogeneous and stable water-dispersible colloidal nanostructures—through an eco-friendly route. Additionally, bearing in mind the potential clinical applications of these designed nanosystems, CMC was selected with a lower degree of substitution (DS = 0.7) to favor biodegradability and a molecular mass (M_w_ = 250,000 Da) above the renal threshold, relevant for passive targeting functionality [31,32]. The relatively high molecular molar therapeutics may preferentially accumulate in tumors to protect polymer–peptide conjugates from rapid blood clearance and elimination from the body, leading to enhanced permeation and retention (EPR) effects [31,32]. Moreover, CMC macromolecules and their hybrid nanostructure derivatives can be internalized in tumoral and non-tumoral cells without additional transmembrane vectors [8]. It should also be mentioned that the choice of Co-doped iron oxide nanoparticles was based on the literature related to the magnetic, chemical, and biological properties of metal-doped ferrites, including the previous study by our group [26], to achieve higher efficiency in chemodynamic (CDT) and magneto-hyperthermia (MHT) therapies. For CDT, incorporating cobalt into a magnetite nanostructure considerably enhances the peroxidase-like activity of Fe_3_O_4_ caused by the higher decomposition of H_2_O_2_ into •OH free radicals. This can be explained based on the redox potential of Co^3+^/Co^2+^ (E^0^ = 1.81 V), which is higher than that of Fe^3+^/Fe^2+^ (E^0^ = 0.77 V). Therefore, the reduction of Co^3+^ species by ferrous ions (Fe^2+^) would be thermodynamically favorable, contributing to the effective regeneration of Co^2+^ and, consequently, increased H_2_O_2_ decomposition. For MHT, Co-doping of iron oxide is an efficient strategy to improve the magnetite nanoparticles’ hyperthermia properties, as it increases the specific absorption rate (SAR), which describes the efficiency of heat conversion. This result is predominantly associated with the increase in the magnetocrystalline anisotropy of the system promoted by the replacement of Fe^2+^ with Co^2+^.

Secondly, AuNP nanozymes were produced by in situ TSC-mediated chemical reduction (AuNP) to ensure biocompatibility combined with colloidal stability in aqueous media. The design choice of trisodium citrate (TSC) as the ligand instead of larger molecules for producing gold-based nanozymes was based on their small size and biocompatibility. This characteristic favors the overall hydrodynamic dimension (D_H_) of the integrated nanostructures to facilitate posterior uptake through endocytosis for killing GBM tumor cells.

Thus, by amalgamating the two components, a dual-nanozyme hybrid nanoarchitecture (AuNP//Co-MION@CMC, dual-nanozyme or Bi-nano) was constructed for cascade biocatalytic activity to kill GBM cancer cells. In the last stage, these nanosystems were biofunctionalized by iRGD (i.e., nine-amino-acid cyclic peptide, sequence: C**RGD**KGPDC) through an EDC-mediated chemical reaction to enable active targeting of GBM cancer cells (AuNP//Co-MION@CMC_iRGD, Bi-nano_iRGD). The conjugation occurred via covalent coupling of the amine groups from iRGD (R-NH_2_) with the carboxylic groups of CMC (R-COOH), forming amide bonds. Thus, this work produced hybrid colloidal biofunctionalized supramolecular nanostructures to be applied as biocatalytic nanovectors for targeting and boosting the deaths of U87 cancer cells (Equations (1) and (2)).
Co-MION@CMC + AuNP → AuNP//Co-MION@CMC(1)
AuNP//Co-MION@CMC + iRGD → AuNP//Co-MION@CMC_iRGD(2)

### 3.1 Material Chemistry Characterization of Nanoconjugate Components

#### 3.1.1. Co-MION@CMC Nanoconjugates

The formation of Cox-MION@CMC nanocolloids was assessed by several characterization techniques to investigate the organic (i.e., carboxymethylcellulose) and inorganic (Cox-MION) components as well as their interactions for resolving the morphological and structural features and properties. To this end, surface chemistry plays a pivotal role in the overall process of the nanoparticles’ formation, stabilization, and potential interactions with the surrounding media. The interactions that developed at the interface between the functional groups of the CMC ligand and the nanoparticles’ surfaces were evaluated by FTIR and high-resolution XPS spectroscopy (HR-XPS).

In Figure 2A, a broad band was observed in both FTIR spectra in the range from 3500 to 3200 cm^−1^, assigned to νO–H vibrations, which are typically present in cellulose and its derivatives, such as CMC-based materials. Additionally, we observed bands associated with asymmetric (1650 and 1590 cm^−1^) and symmetric (1420 and 1320 cm^−1^) stretching of COO^−^ groups related to the carboxylate functionalities of CMC, as well as the bands assigned to carboxylic species (νC=O, 1730 cm^−1^ and νC-O, 1250). According to the literature, two bands of symmetric and asymmetric carboxylates indicate two different types of coordination in polymer–metal complexes. Moreover, vibrations from secondary alcohols (νC2–OH; 1110 cm^−1^; νC3–OH; 1060 cm^−1^) and β1-4 glycoside bonds at 890 cm^−1^ were detected [33]. Furthermore, bands centered at 620–590 cm^−1^ were assigned to stretching vibrations of the tetrahedral group of the metal ion–oxygen complex (M–O) [34]. The XPS results for the organic shell components indicated the chemical states of carbon (Figure 2B) and oxygen (Figure 2C) assigned to the CMC polymer [33,35] being “free” in solution (a) and acting as a capping ligand of iron oxide nanoparticles (b). The changes in the areas associated with each chemical bond and the shift in binding energy (C=O, Δ = 1.1 eV) could be related to the occurrence of interactions between Fe and Co and the functional groups of the polymer at nanointerfaces, resulting in the rearrangement of the conformational structure of the polymer upon nanoparticle stabilization. In the spectrum of Co-MION@CMC, in the O 1s region (Figure 2C(b)), the chemical state of the M–O bond (530.6 eV) was also observed [36,37].

The colloidal properties of Co-MION@CMC were evaluated by hydrodynamic diameter (D_H_) determined by dynamic light scattering (DLS) and zeta potential (ZP). A zeta potential value of −41 ± 5 mV evidenced the anionic characteristics of the carboxylate functional groups from CMC, which were expected based on its pKa (~4.3), rendering stability to the nanocolloids (Co-MION@CMC) primarily through electrostatic repulsion, but also through steric hindrance. The average D_H_ was 45 ± 4 nm, corresponding to the sum of the inorganic core (i.e., Co-doped iron oxide nanoparticles) and the CMC shell “swollen” in the aqueous medium (polydispersity index (PDI): 0.258). The schematic representation of the Co-MION@CMC is depicted in Figure 2D.

Regarding the inorganic core, HR-XPS analysis of the Fe 2p region (Figure 2E) revealed the characteristic peaks of magnetite at binding energies of 710.4 eV and 723.8 eV, overlapping the contributions of Fe^2+^ and Fe^3+^, where the absence of a satellite peak at 719 eV confirmed both ferrous and ferric species in the lattice. The HR-XPS spectrum of the Co 2p region (Figure 2F) presented the spin–orbit peaks at 781.1 eV (Co 2p_3/2_) and 797.1 eV (Co 2p_1/2_). The shake-up satellite peaks (786.1 eV, Co 2p_3/2_, and 803.1 eV, Co 2p_1/2_) were consistent with the presence of the Co^2+^ chemical state [26,38].

The X-ray diffraction (XRD) pattern of Co-MION@CMC indicated that the nanoparticles were crystalline, with an inverse spinel structure of magnetite iron oxide (ICCD, The International Centre for Diffraction Data, file 89-0691), despite the doping. Diffraction peaks were observed at 2 theta equal to 30.1°, 35.2°, 43.2°, 57.1°, and 62.7°, associated with the (220), (311), (400), (511), and (440) planes, respectively (Figure 2G). The average nanocrystallite size of 7.2 ± 0.7 nm estimated based on the Scherrer equation using (311) reflection was consistent with the nanoparticles’ average size (diameter = 6.8 ± 1.1 nm) verified via TEM images (Figure 2H). Moreover, the transmission electron microscopy (TEM) analysis indicated that the Co-MION@CMC nanoparticles were monodisperse (PDI = 0.026) with spherical morphology. The localized diffraction of the electron patterns (SAED, inset Figure 2G) presented the interplanar lattice distances (±0.1 Å) of 3.0 Å, 2.5 Å, 2.1 Å, 1.6 Å, and 1.5 Å, consistent with the XRD analysis of magnetite (diffraction of the (220), (311), (400), (511), and (440) planes, respectively). The content of cobalt, estimated by WD-XRF, was 3.1 mol% (relative to the total iron content), consistent with the theoretical amount (3.3%) of synthesis.

Hence, the first of the chief hypotheses of this research was demonstrated—that water-dispersible colloidal nanoconjugates could be produced based on a Co-MION core and CMC shell for applications as nanozymes in the designed biocatalytic cascades.

#### 3.1.2. AuNP Nanoconjugates

As the second subunit of the designed hybrid nanoassemblies, colloidal gold nanoparticles stabilized by trisodium citrate (TSC) were comprehensively characterized to assess their physicochemical properties before being applied as oxidase-like nanozymes in the biocatalytic application. Thus, the UV–Vis spectrum of the AuNPs is presented in Figure 3A, where a significant absorption band at 520 nm can be observed, associated with the localized surface plasmon resonance (LSPR) phenomenon. This result demonstrates the effective formation and stabilization of metallic gold nanoparticles by citrate through in situ reduction. The reduction of Au^3+^ species in solution to Au^0^ nanoparticles was also clearly noted based on the reddish-purple color of the suspension.

The FTIR spectrum of TSC-capped AuNPs is presented in Figure 3B. It indicates peaks of the ν_as_ COO^−^ (1605–1555 cm^−1^) and ν_s_ COO^−^ (1405–1370 cm^−1^), which are characteristic of citrate species in TSC-stabilized gold nanoparticles. The broadening of the band of asymmetric carboxylate vibration at higher wavenumbers is associated with the bending vibrations of adsorbed water (δ OH) on metal surfaces, usually detected in the range of 1650–1610 cm^−1^ [39]. Regarding the colloidal physicochemical properties, the D_H_ was evaluated as 15 ± 1 nm, with a PDI of 0.250. A zeta potential value of −50 ± 5 mV indicated that electrostatic forces effectively stabilized the AuNPs due to the repulsion between anionic charges from COO^−^ groups from citrate (ZP < −30 mV) (inset in Figure 3B). The TEM image (Figure 3C) indicated that the AuNPs presented a spherical shape and monodispersed size distribution (PDI = 0.035) centered at 16 ± 3 nm (Figure 3D), consistent with the size measured by DLS. This was expected, considering that the diameter obtained by laser scattering for TSC-AuNPs is usually similar to that obtained by TEM, due to the low contribution of the small molecules in the organic layer. The crystalline nature and composition of the nanoparticles were verified by SAED (Figure 3E). The concentric diffraction rings corresponded to interplanar distances of 2.4 ± 0.1 Å, 2.0 ± 0.1 Å, 1.4 ± 0.1 Å, and 1.2 ± 0.1 Å, which were indexed as the (111), (200), (220), and (311) planes of the face-centered cubic (FCC) nanostructure of Au^0^ (ICDD file 04-0784), respectively. HR-XPS analysis (Figure 3F) also confirmed the absence of Au^3+^ species and the formation of metallic gold. The Au 4f region (Au 4f_7/2_ and Au 4f_5/2_) was deconvoluted into two peaks compatible with low-coordinate Au^0^ species at the surface (83.3 eV and 87.0 eV) and bulk Au^0^ (84.0 eV and 87.7 eV) [40].

At this point, the second of the chief hypotheses of this research has been demonstrated, where water-dispersible colloidal nanoconjugates were produced based on a gold nanoparticle core and citrate shell for applications as nanozymes in the biocatalytic cascades.

#### 3.1.3. Dual-Nanozyme Nanoassemblies

UV–Vis characterization of the dual-nanozyme nanoassemblies (Figure 4A) indicated that the spectrum of the combined suspension retained the typical absorption signatures of the individual components—AuNPs and Co-MION@CMC. A *blue-shift* was observed (from 520 to 512 nm) associated with the LSPR peak, indicating the coupling between these nanozymes. Digital images of the nanozymes and dual-nanozyme suspensions indicate the colloidal nanomaterials’ stability before and after the coupling (Figure 4B).

Annular dark-field imaging, in comparison to the TEM image (Figure 4C), was used to depict both nanozyme components (i.e., AuNPs and Co-MION) in the dual-nanozyme assemblies (Bi-nano), because this technique is highly sensitive to the atomic number (Z) of elements. As the atomic numbers of iron and oxygen (26 and 8, respectively) are lower than that of Au (Z = 79), the brighter areas in the image are associated with gold nanoparticles. This was expected based on the differences in the sizes of the nanomaterials verified in the previous analysis (Figure 2 and Figure 3). In addition, TEM images of the hybrids at higher magnifications (Figure 4D) showed AuNPs and Co-doped iron oxide nanoparticles where no significant changes were detected in the nanoparticles’ size and morphology. EDS spectra focused on each nanozyme (see the EDS spots in Appendix A) indicated the presence of Fe, O, and Co (low signal of Co due to small content in the sample: Co^2+^ < 1.5 mol% relative to the iron oxide) associated with regions with predominant Co-MION nanoparticles. For the EDS spectrum collected at gold-nanoparticle-rich areas, Au was detected with only a small content of iron due to the proximity of Co-MION near the e-beam spot of analysis. Additionally, in both EDS spectra, C and O were detected from the nanoparticles’ capping ligands. Cu and Si were related to the grid used as support for sample deposition and the microscope detector, respectively.

Zeta potential analysis and DLS were employed to assess the surface charge and hydrodynamic evolution of the bifunctional nanozymes upon the coupling process. Upon coupling, the ZP of the nanosystem was −45 ± 4 mV, similar to that obtained for the Co-doped nanostructure (−41 ± 5 mV), consistent with the formation of the inorganic–inorganic dual-nanozyme system functionalized by a CMC shell. DLS measurements also supported the effective formation of the amalgamated nanosystems dispersed in the aqueous medium, as the D_H_ for the dual-nanozyme system was 28 ± 3 nm—an intermediate value between those obtained for the individual components. This result can be observed from the normalized correlation curves shown in Figure 4E. The observed “shifts” of values of time (τ) unequivocally demonstrated the effective interactions between Co-MION@CMC (Figure 4E(a)) and AuNPs (Figure 4E(b)) caused by the change in inertia of the coupled systems (Figure 4E(c)).

The FTIR spectrum of the AuNPs (Figure 4F(b) and Figure 3B) demonstrated the citrate groups used as chemical reductants and stabilizing agents for gold nanoparticles. The spectrum of Co-doped magnetite presented the main groups of the CMC polymer (Figure 4F(a) and Figure 2A). Upon combining both nanozymes (Figure 4F(c)), a significant change was detected in the vibrations of bands associated with hydroxyl groups/hydrogen bonds, indicating the change in the supramolecular structure of the polymer in the nanoassembly.

### 3.2. Catalytic Activity of Nanoconjugates—In Vitro Acellular Analysis

#### 3.2.1. Co-MION@CMC Nanozyme Activity

Considering the more acidic pH of the tumor microenvironment (TME), the catalytic activity of the Co-MION nanozyme was compared under mildly acidic (pH = 5.0) and neutral (pH = 7.0) conditions, which are characteristics of cancer and healthy cells, respectively (Figure 5A). Hence, based on TMB_ox_ formation, as the typical chromogenic model for catalytic analysis, it was observed that Co-MION@CMC presented peroxidase-like activity at pH 5.0 (Equation (3)), whereas at pH 7.0 the behavior was primarily catalase-like (Equation (4)). The results were also dependent on substrate concentration (Figure 5A), the concentration of the nanozyme (Appendix A), and temperature (Appendix A). Thus, these results confirmed that the produced Co-MION@CMC nanozyme could offer the opportunity for catalytic-based therapy based on the pH-dependent response, which is more active in cancer cells, i.e., under acidic tumor conditions. Consequently, it can boost the formation of radical species (ROS) for fighting cancer cells while reducing potential side effects on healthy cells (i.e., at neutral pH), tissues, and organs.
(3)$$H_{2}O_{2}\overset{\mathrm{Co}-\mathrm{MION}@\mathrm{CMC}}{\to }•{\mathrm{OH}+\mathrm{OH}}^{-}$$
(4)$${2H}_{2}O_{2}\overset{\mathrm{Co}-\mathrm{MION}@\mathrm{CMC}}{\to }{2H}_{2}O+{O_{2}}^{-}$$

#### 3.2.2. AuNP Nanozyme Catalytic Activity

The capability of AuNP@TSC to generate hydrogen peroxide upon glucose injection was evaluated in the presence of HRP and TMB according to Equation (5) and Equation (6):
AuNPβ-D-glucose + O_2_ + H_2_O   →   β-D-gluconic acid + H_2_O_2_(5)
HRPH_2_O_2_ + TMB  →  TMB_ox_ + H_2_O(6)

The results in Figure 5B demonstrate the catalytic activity of the AuNPs when submitted to the β-d-glucose substrate and its dependence on pH. As the HRP activity is also dependent on pH (i.e., higher at pH = 7.0 than pH = 5.0), the catalytic behavior of the AuNPs was more pronounced at neutral pH. This assigned the predominantly oxidase-like catalytic activity to the AuNP nanozyme—essential for in situ production of H_2_O_2_ to enable the biocatalytic cascade using intracellular glucose as the substrate source.

#### 3.2.3. Dual-Nanozyme AuNP//Co-MION@CMC Catalytic Cascade Activity

**Assay of subunits of cascade components:** As demonstrated in the previous sections, both nanoconjugates showed nanozyme activity when tested separately. Next, upon coupling for producing the dual-nanozyme nanoassemblies, the results demonstrated that each nanozyme retained its original catalytic activity when assayed using their specific substrates (i.e., β-D-glucose for AuNP and H_2_O_2_ for Co-MION@CMC) (Figure 5C).

**Assay of dual-nanozyme catalytic cascade activity:** The inorganic dual-nanozyme cascade integrating AuNPs and Co-MION@CMC was assayed upon injecting the glucose (substrate of AuNPs) to evaluate the catalytic cascade activity for oxidation of TMB. The results (Figure 5D) demonstrated the formation of TMB_ox_ following the two-stage cascade reactions summarized in Figure 5E: (I) the substrate (β-D-glucose) in the presence of O_2_ (dissolved in medium) and gold nanoparticles (as the first nanozyme, GOD) produced H_2_O_2_ and gluconic acid, and (II) the hydrogen peroxide in the presence of Co-MION@CMC (second nanozyme, POD) generated hydroxyl radicals (•OH) from the disproportionation of the H_2_O_2_ oxidizing TMB to TMB_ox_. The evolution of the oxidation of TMB over time at two concentrations of the initial substrate of the cascade (glucose) illustrates the overall cascade behavior, demonstrating the feasibility and reactivity of the inorganic multi-enzymatic reactions in vitro (inset, Figure 5E).

Although the Michaelis–Menten (MM) equation could be used to reveal important information regarding the efficiency of catalysis for most natural enzymes and for analyzing nanozymes, when considering biocatalytic cascades the behavior is usually far more complex. Therefore, in addition to MM analysis for defining the optimized running conditions of each nanosystem, several steps of modeling/refining the parameters of the whole reaction system are required to achieve efficient catalysis, which was not the focus of this paper [41,42].

### 3.3. Biocatalytic Assay In Vitro with Live Cell Cultures

#### 3.3.1. Intracellular ROS Formation—Nanozyme-Induced Biocatalytic Activity

As the catalytic activity of nanozymes has been associated with the generation of reactive oxygen species (ROS), experiments involving the detection and quantification of intracellular ROS formation were designed and performed. The reactive oxygen species measurements were executed using the staining of these species with 5-(and -6)-carboxy-2′,7′-dichlorodihydrofluorescein diacetate (DCF-DA), based on photoluminescence spectroscopy (PL). Essentially, DCF-DA (non-fluorescent)—a cell-membrane-permeable probe—diffuses into the cells, where it becomes hydrolyzed to DCFH (dichlorodihydrofluorescein, non-fluorescent), which remains trapped within the cell. After the cells take up the active nanozymes, DCFH reacts with the H_2_O_2_, hydroxyls, and peroxyl radicals generated intracellularly, producing the fluorescent 2′,7′-dichlorofluorescein (DCF), thereby enabling the estimation of these ROS species via the fluorescence emission intensity of DCF [43]. The schematic representation of the bioassay and the results of ROS detection based on the PL intensity of each nanosystem incubated with the cell cultures are presented in Figure 6A.

The biocatalytic activity of the AuNP and Co-MION@CMC inorganic nanozymes presented a similar trend to that observed for the catalytic behavior of the previous acellular results (i.e., without cells). The PL intensity of the intracellular ROS accumulation in the presence of AuNPs was ascribed to the catalysis of endogenous glucose to generate H_2_O_2_ and gluconic acid in the presence of O_2_, mimicking glucose oxidase (GOD) enzyme-like activity (Figure 6B). For the Co-MION@CMC nanozyme system, the fluorescence intensity relies on the reaction involving the consumption of H_2_O_2_. This is naturally present inside cells, but is also produced due to the formation of hydroxyl radicals (•OH) via a Fenton-like reaction compatible with a peroxidase-like (POD) behavior (Figure 6C). Considering the amalgamated catalytic cascade forming the dual-nanozyme nanoassemblies, the overall balance of the biocatalytic activity increased the amounts of ROS inside the cell compared to each component separately (Figure 6D,E). ROS formation generally tended to increase over time for all systems, but it was also dependent on nanozyme concentration. It was also influenced by cell type (Figure 6F), where higher amounts of ROS were detected for the glioblastoma cell line (U87) compared to healthy cells (HEK 293T). These findings were ascribed to the enhanced metabolism and the higher levels of H_2_O_2_ in cancer cells. The higher kinetics of cancer cells endorsed these results even at the early stages (15 min), where approximately 95% of the ROS signal was already detected, compared to only 50% in the healthy cells. Thus, this trend offers an exciting opportunity for a *“drug-free”* chemotherapy treatment relying strictly on the biocatalytic activity of dual-nanozyme systems and the distinct intrinsic metabolism of cancer cells compared to healthy cells.

#### 3.3.2. MTT Cytotoxicity—Cell Viability Assay

In addition to the ROS generation experiments, the MTT bioassay was used to evaluate the potential cytotoxicity of the nanosystems, which is highly desired for killing cancer cells but should be avoided with healthy cells. In general, the results (Figure 7A) indicated that the cell viability responses were affected by the concentrations of the nanozymes, the specificity of the individual components (i.e., AuNPs and Co-MION@CMC), and the combination of the systems into nanoassemblies (AuNP//Co-MION@CMC). This trend was assigned to the biocatalytic activity of the nanozymes, which confirmed the hypotheses that each nanozyme and their amalgamation after the cell’s uptake affected the cellular metabolism, leading to intensification of cytotoxicity as the concentration was increased.

As the ROS species produced by AuNP nanoconjugates have relatively lower reactivity (primarily H_2_O_2_) [44], the cell death after exposure to this nanozyme is expected to be mostly related to *starvation* due to glucose depletion caused by the OD-like catalytic activity of gold nanoparticles (Equation (5)).

On the other hand, for Co-MION@CMC—a magnetite-based inorganic core—it is well-established that iron oxide nanostructures have great potential as peroxidase-like (POD) nanozymes for chemodynamic therapy (CDT) [44,45]. The imbalance of reactive oxygen species generated by these nanozymes often causes oxidative stress that engenders intracellular damage to cells. This is the basis for chemodynamic therapy using ROS-modulating agents that elevate the levels of highly reactive radicals to inhibit growth and eliminate cancer tumor cells. Hence, once the Co-MION@CMC nanoconjugates have undergone cellular uptake, they convert the less-reactive endogenous H_2_O_2_ to highly harmful hydroxyl (•OH) and hydroperoxyl (•HO_2_) radicals (Equation (7) and Equation (8)), promoting enhanced Fenton-like ROS damage. Furthermore, treatment-facilitative effects and cell damage mechanisms of iron-based nanostructures also involve the activation of cell death signaling by the ferroptosis pathway associated with lipid peroxidation [44,45].
Fe^2 +^_(surface)_ + H_2_O_2(ads)_ → Fe^3 +^_(surface)_ + •OH _(intracellular)_ + OH^−^_(intracellular)_(7)
Fe^3 +^_(surface)_ + H_2_O_2(ads)_ → Fe^2 +^_(surface)_ + •HO_2 (intracellular)_ + H^+^_(intracellular)_(8)

Although overproduction of H_2_O_2_ is often attributed to the tumor microenvironment (TME) [45], the original strategy adopted in this research using the combination with AuNP oxidase-mimicking nanozymes to produce the dual-nanozyme nanoassemblies (i.e., binanozymes) was used to increase the intratumoral hydrogen peroxide (ΔH_2_O_2_) concentration based on the biocatalytic cascade reactions presented in Figure 5E, enhancing the cell death shown in Figure 7A.

When considering the effect of cell type (Figure 7B), the results evidenced relatively higher cytotoxicity of the dual nanozymes (AuNP//Co-MION@CMC) towards cancer cells (U87) compared to healthy cells (Δ~15%). It should be stressed that this result indicates specificity toward killing tumor cells as opposed to healthy cells, which is a vital feature of nanomedicines for prospective applications in cancer theranostics. As discussed in the previous section with respect to the ROS bioassay, this specificity can be mainly credited to the different metabolic activities and pathways of cancerous cells, which are more intense than those of normal cells. Because glucose consumption—as the primary energy source (i.e., *“biofuel”*) and the substrate molecule of the AuNP nanozyme catalyst—is intensified by cancer cells, it generates higher intracellular stress of toxic species, boosting cell death.

#### 3.3.3. Nanozyme-Triggered Cell Death by Ferroptosis

To further investigate the cell death pathways, the lipid peroxidation process was assessed. Due to their high contents of polyunsaturated fatty acids, cellular and organelle membranes are especially susceptible to ROS damage, termed lipid peroxidation (LiP). This is a process in which free ROS remove electrons from lipids and produce new reactive intermediates in a chain reaction. The lipid peroxidation process damages phospholipids by directly altering their structure, activity, and physical properties, which can induce cell death signaling known as “ferroptosis”. More specifically, ferroptosis occurs due to increased ROS presence provoked by high intracellular levels of iron species and the depletion of the endogenous antioxidant glutathione (GSH). The most accepted mechanism involves the redox couple, where GSH is oxidized into glutathione disulfide (GSSH) by the ROS species, reducing the intracellular levels of hydrogen peroxide and phospholipid hydroperoxides (LP-OOH), and causing lipid peroxidation. Thus, this mechanism leads to programmed cell death, i.e., ferroptosis, which is distinct from the apoptotic mechanism [44,45,46]. In this sense, lipid damage was evaluated by estimating levels of malondialdehyde (MDA)—one of the products of the peroxidation process of polyunsaturated fatty acids—and it was measured using the thiobarbituric acid (TBA) protocol as an indicator of oxidative stress/lipid peroxidation. The results shown in Figure 8A indicate that the MDA content was significantly increased after incubation with both nanozymes and the binanozyme system compared to the negative control (one-way ANOVA and Bonferroni’s test, *p* < 0.0001). These findings corroborate the formation of oxidative species promoted by the nanozymes after cellular uptake. At lower concentrations of both nanozymes, there was practically no generation of the MDA lipid peroxidation marker. Conversely, a more prominent intensity of the MDA biomarker was detected with the biocatalytic cascade and Co-MION@CMC at higher concentrations. Comparing the results of MDA (i.e., LiP) with cell viability responses (i.e., MTT), it can be suggested that two major mechanisms associated with cell death are taking place (Figure 8B,C). In Region-(I) of the plot, at MDA concentrations of 1.0 nmol/mg, the correspondent toxicity observed through the cell viability assay was below 55%, which could be primarily credited to the apoptosis mechanism caused by ROS. For Region-(II) (MDA > 1.5 nmol/mg), both apoptosis and ferroptosis mechanisms were expected to occur due to the higher values of oxidative species/lipid peroxidation and the presence of iron ions in these nanosystems (Co-MION@CMC and binanozyme), which are characteristic of the ferroptosis cell death mechanism. It has also been reported that MDA combines with DNA, which can activate signal transduction pathways, inducing apoptosis and, ultimately, resulting in cell death [47]. As the dysfunction in lipid metabolism is a hallmark of cancer cells, the ferroptosis-triggered cell death caused by lipid peroxidation is a vital opportunity for using nanozymes in novel cancer therapy [48,49,50].

#### 3.3.4. Cancer Cell Targeting via Membrane Integrin Receptors— iRGD

As active targeting is crucial in cancer therapy, the iRGD targeting peptide (Cys-Arg-Gly-Asp-Lys-Gly-Pro-Asp-Cys; disulfide bridge: Cys1-Cys9) was grafted to the nanoassemblies to improve their selectivity towards brain cancer cells. In addition, iRGD has been demonstrated to promote the effective transportation of nanostructures through the blood–brain barrier (BBB), which is vital for the success of brain cancer therapy [51]. Thus, it was conjugated to the dual-nanozyme supramolecular nanosystems using EDC chemistry. EDC is a water-soluble, “zero-length” crosslinker that is not incorporated into the conjugated structure by the chemical reaction; therefore, there are no concerns about its toxicity compared to other crosslinkers (e.g., glutaraldehyde, formaldehyde, epichlorohydrin) [52]. Thus, the biofunctionalized AuNP//Co-MION@CMC-iRGD (Bi-nano_iRGD) colloidal water-dispersible nanoassemblies were extensively characterized and verified for targeting brain cancer cells under a multiple-therapy approach.

##### Material Chemistry Characterization of Bi-Nano_iRGD

The conjugation of CMC in the dual-nanozyme system with iRGD was confirmed by FTIR spectroscopy (Figure 9A), where the major bands associated with the functional groups of the carboxymethylcellulose biopolymer (e.g., carboxylates) were observed as previously described in Section 3.1.1. In addition, after the EDC-mediated conjugation, the bands related to amide groups (-N(H)-C(=O)) were observed at 1655 (amide I, νC=O), 1540 (amide II, δN-H), and 1240 (amide III, νC-N) cm^−1^ [8]. These were associated with the conjugation through covalent bonds between carboxylic groups of CMC and amino groups of iRGD, but also due to the peptide bonds of the nine-amino-acid cyclic sequence of the active targeting biomolecule.

Furthermore, the grafting of iRGD to the dual-nanozyme system was also established based on changes in the surface charges and supramolecular nanostructures (Figure 9B). Upon conjugation, the ZP value was significantly altered from −45 ± 4 mV to −29 ± 3 mV (Δ = ~35%). This effect was predominantly attributed to the relative consumption of carboxylate groups from the anionic polymer to yield amide covalent bonds, considering that the projected net charge contribution of the iRGD peptide sequence was approximately zero (isoelectric point = 7.07; PepCalc.com, 2015, Innovagen AB©). Moreover, the initial hydrodynamic diameter of the dual-nanozyme system (D_H_ = 28 ± 3 nm) showed a significant increase (Δ = ~100%, to 60 ± 4 nm) after the chemical conjugation with iRGD. This effect was credited to the changes in the 3D conformation of the functionalized supramolecular nanoassemblies dispersed in the aqueous medium by introducing the nine-amino-acid cyclic peptide, affecting the original overall balance of hydrophilic/hydrophobic interactions, as verified by ZP results.

##### Peptide-Targeted Biocatalytic In Vitro Assay

The results demonstrated that the biofunctionalization with iRGD targeting peptides remarkably boosted the cell-killing activity of the dual-nanozyme assemblies toward brain cancer cells (Δ~ +36%, Figure 9C). This effect was attributed to the tumor-homing of the cell-targeting peptides, predominantly driven by their high affinity for α_v_β_3_ membrane integrin receptors [8,33]. More importantly, these iRGD biofunctionalized nanovectors demonstrated more prominent cell-killing activity of approximately 60% toward cancer cells compared to healthy cells (HEK 293T, ~23%, inset Figure 9C). This outcome was ascribed to the combination of the targeting characteristics of the iRGD peptides for reaching membrane integrin receptors overexpressed by U87 cancer cells with the increased cellular uptake. Although RGD-based sequences are not specific cell-penetrating peptides according to the literature [8,33], the affinity to membrane integrin receptors usually induces higher cellular uptake of biofunctionalized nanosystems, as can be observed via bioimaging methods.

The ROS production results of U87 cells depicted in Figure 9D for iRGD-targeted and non-targeted nanozymes (AuNP//Co-MION@CMC) indicate that, in the early stages (i.e., 15 min), the cell uptake kinetics were very rapid and had similar values. However, as the incubation time increased, after 60 min, it appeared that the alternative pathway usually found in cancer cells to hamper the excess generation of intracellular ROS was hampered. Consequently, the iRGD-modified nanoassemblies promoted a relatively higher increase in ROS in the cell compartment than the non-functionalized systems. On the other hand, as expected, the healthy cells (HEK 293T) presented a much lower generation of ROS than cancer cells for both functionalized and non-functionalized nanoassemblies. In addition, no significant difference in ROS production between the functionalized and non-modified nanosystems was observed with healthy cells. This could be associated with a much lower number of integrin receptors at the cell membrane and a slower metabolism.

#### 3.3.5. Multimodal Brain-Cancer-Targeted Therapy—Biocatalytic–Magnetothermal–Chemotherapy

**Magnetic properties of Co-MION@CMC:** Initially, the magnetic properties of iron-oxide-based nanoparticles (i.e., Co-MION@CMC) were assessed by vibrating sample magnetometry (VSM), aiming to produce magnetically responsive nanosystems. The magnetization *versus* magnetic field for Co-MION@CMC nanoparticles at room temperature indicated superparamagnetic behavior due to the absence of the hysteresis loop, which characterizes the complete reversibility of the magnetization process (Figure 10A). Moreover, to further characterize the magneto-responsive behavior of the nanoconjugates for potential magnetic hyperthermia applications (MHT, a heat-damage-induced anticancer therapy), the Co-MION@CMC colloidal solution was submitted to an external alternating magnetic field (AMF, before cell incubation). After 30 min under the AMF (frequency = 112.6 kHz; amplitude = 19.9 kA/m), the colloidal temperature increased by 5.2 ± 1.9 °C, and the calculated specific absorption rate (SAR)—which describes the efficiency of heat conversion—was 12 ± 3 W/g metal (Figure 10B). Thus, one of the primary hypotheses of this research was demonstrated, where stable water-dispersible colloidal nanostructures were produced based on a carboxymethylcellulose macromolecular shell and superparamagnetic Co_0.1_Fe_2.9_O_4_ nanoparticle core, suitable for magnetically responsive nanosystems in hyperthermal applications.

Based on these findings, it was projected that after delivering the magnetic nanoassemblies to U87 tumor cells, the submission to local magnetic hyperthermal therapy (MHT) would be able to increase the temperature in the surrounding medium by approximately 5 °C. Theoretically, considering the normal human body temperature, this temperature rise would be appropriate to induce cell death by apoptosis and necrosis [26,53]. It is vital to highlight that the H.f value obtained under these conditions (2.2 x 10^9^ A/m.s) was lower than the recommended limit dose (5 ×10^9^ A/m·s), considering the safety of the patient under therapy [54].

**Biocatalytic–Magnetothermal Targeted Therapy with Brain Cancer Cells in vitro:** We evaluated the effects of magnetothermal therapy after incubation for 3 h with dual-nanozyme integrated systems, with and without targeting—i.e., Bi-nano_iRGD and Bi-nano, respectively—where U87 tumor cells were exposed to an alternating magnetic field (19.9 kA/m) for 60 min.

The cell viability responses were 26% and 8% for the Bi-nano_MHT and Bi-nano_iRGD_MHT nanosystems, respectively (Figure 10C). These findings indicated a significant reduction of 35% and 65% in the cell viability, respectively (i.e., equivalent to higher cytotoxicity), compared to “control” samples in the absence of the magnetic hyperthermia treatment (cell viability of 40% for Bi-nano and 23% for Bi-nano_iRGD). Moreover, the killing activity was superior in peptide-targeted dual nanozymes, which is highly desirable when building anticancer nanocarriers. Thus, these results validated the MHT as an effective additional treatment for killing cells due to highly localized heat generated by the nanoassemblies—primarily due to the Co-MION@CMC nanozyme component—when submitted to an external alternating magnetic field. The thermal imaging technique has been reported [55] as a supporting method for the localized analysis of heat generated by MHT in cells and tissues, but it is challenging and requires extremely sophisticated apparatus and equipment. These results are also consistent with the development of a dual-mode therapeutic strategy based on the biocatalytic behavior of the nanozyme cascade (i.e., chemodynamic therapy, CDT) and the magnetothermal properties of Co-doped magnetite nanoparticles with enhanced efficiency in killing brain cancer cells due to their peptide-targeted functionalization effect. It should be highlighted that, in future studies, numerous bioimaging techniques can be used to corroborate these findings while offering complementary information regarding extra- and intracellular interactions at biointerfaces with these dual-nanozyme nanoassemblies [7,8,9,18,55,56].

The complete schematic representation of the dual-nanozyme supramolecular assemblies’ mechanisms and pathways for the multimodal magnetothermal–chemodynamic targeted therapy against brain cancer cells designed and produced in this work is depicted in Figure 11.

### 3.4. Stability and Fate of Nanosystems

Regarding the use of nanoparticles and nanostructures for future clinical applications in theranostics, two critical conditions should be considered: the first is the long-term stability of the nanomaterials (i.e., “shelf-life”), and the other is the fate (i.e., biodistribution, mechanisms of clearance, and excretion pathways of nanoparticles from the living system) of the nanosystems after administration.

#### 3.4.1. Stability of Nanozymes

Bearing in mind the long-term stability evaluation, the nanozymes were stored at 6 ± 2˚C and protected from light for over 12 months after synthesis. After this time, stability was evaluated qualitatively and quantitatively. Upon visual inspection, the suspensions remained homogeneous, without the presence of aggregates or precipitates. Quantitatively, the use of diameter measured by TEM confirmed the stability of the inorganic core (Appendix A; one-way ANOVA, α = 0.05). Furthermore, D_H_ and ZP remained statistically unchanged (Appendix A; one-way ANOVA, α = 0.05). For dual-nanozyme assemblies, as they were always freshly prepared before each set of experiments, the colloidal stability was not an essential aspect.

#### 3.4.2. Fate of the Dual-Nanozyme Assemblies—A Theoretical Approach

There are several challenges involving the transition from the laboratory to the clinical use of nanomaterials, and important factors include the biodistribution and clearance/excretion of the nanoformulations. Their dependence on the administration route and physicochemical properties (e.g., size, shape, surface charge, and functionalization) is well-established in the literature. Once in the bloodstream, the nanoparticles are distributed to various organs (e.g., liver, spleen, kidneys, heart, lungs, brain, etc.). According to the literature [57,58,59,60], the liver and spleen have been identified as the organs of living systems with the greatest nanoparticle accumulation. This trend tends to be enhanced by increasing the nanoparticles’ hydrodynamic diameter. Essentially, when nanoparticles are typically below 10 nm in diameter, they usually accumulate in the kidneys, with higher chances of being eliminated through renal filtration. Clearance of nanoformulations occurs via urine and feces, with the concentration decaying over time. In this sense, to improve systemic circulation and reduce the reticuloendothelial system (RES) interaction favoring the use of the nanoparticles for cancer treatment, nanosystems’ features could be designed, including hydrodynamic diameter, zeta potential, capping ligand/surface chemistry, and functionalization by biomolecules. Based on the size of the nanoparticles and nanosystems developed in this work (D_H_, 15 nm–60 nm), they will most likely evade renal clearance but could be captured by the liver and spleen. In addition, their size is also appropriated for tumor uptake. The presence of CMC on the surface renders a predominantly negative charge (−29 mV to −50 mV) that hypothetically favors capture by the RES, which tends to decrease with the reduction in surface charge. For iRGD-modified nanostructures, the lower ZP is a property that could favor longer blood circulation times which, summed to their targeting functionality, would be expected to facilitate their theranostic activity and reduce side effects on healthy cells/tissues. It should be highlighted that this discussion is presented as a preliminary predictive analysis based on the results and supported by scientific evidence from the literature. Still, it will need to be researched in future studies for validation.

## 4. Conclusions

In summary, in this study, a toxic-drug-free biocatalytic–thermal tumor therapy was developed based on dual-inorganic nanoscale enzyme-mimics composed of cobalt-doped magnetite and gold nanoparticles loaded in supramolecular CMC biopolymer nanostructures. The catalytic treatment relies on the cascade reaction triggered by the oxidase-like activity of gold nanoparticles that uses endogenous glucose to produce H_2_O_2_, which is the substrate for the generation of highly oxidizing reactive radicals by Co-Fe_3_O_4_ through Fenton-like reactions. The cell death induced by heat depends on magnetic hyperthermia of Co-Fe_3_O_4_ nanozymes upon exposure to an external alternating magnetic field. The CMC polymer, which was used to directly stabilize Co-Fe_3_O_4_ nanoparticles and then amalgamated to AuNPs stabilized by TSC, also renders water solubility, biocompatibility, enhanced permeability and retention (EPR), and carboxylic groups for coupling multiple functionalities to the nanoplatform. In this case, the COO^−^ of the biopolymer was used to covalently attach the iRGD sequence as a targeting peptide able to identify ∝_v_β_3_ integrins, which are usually overexpressed in brain tumor cell membranes. Based on this work, it can be envisioned that such an approach offers novel multimodal nanoplatforms for treating glioblastoma, which remains the most common and aggressive primary tumor in the adult brain, where the current standard of cancer treatment relies on surgical excision of the tumor followed by ionizing radiation (IR) and chemotherapy. However, despite all of these intense treatments with often severe side effects, the median survival rate for patients remains less than 2 years.

## Figures and Tables

**Figure 1 pharmaceutics-14-02223-f001:**
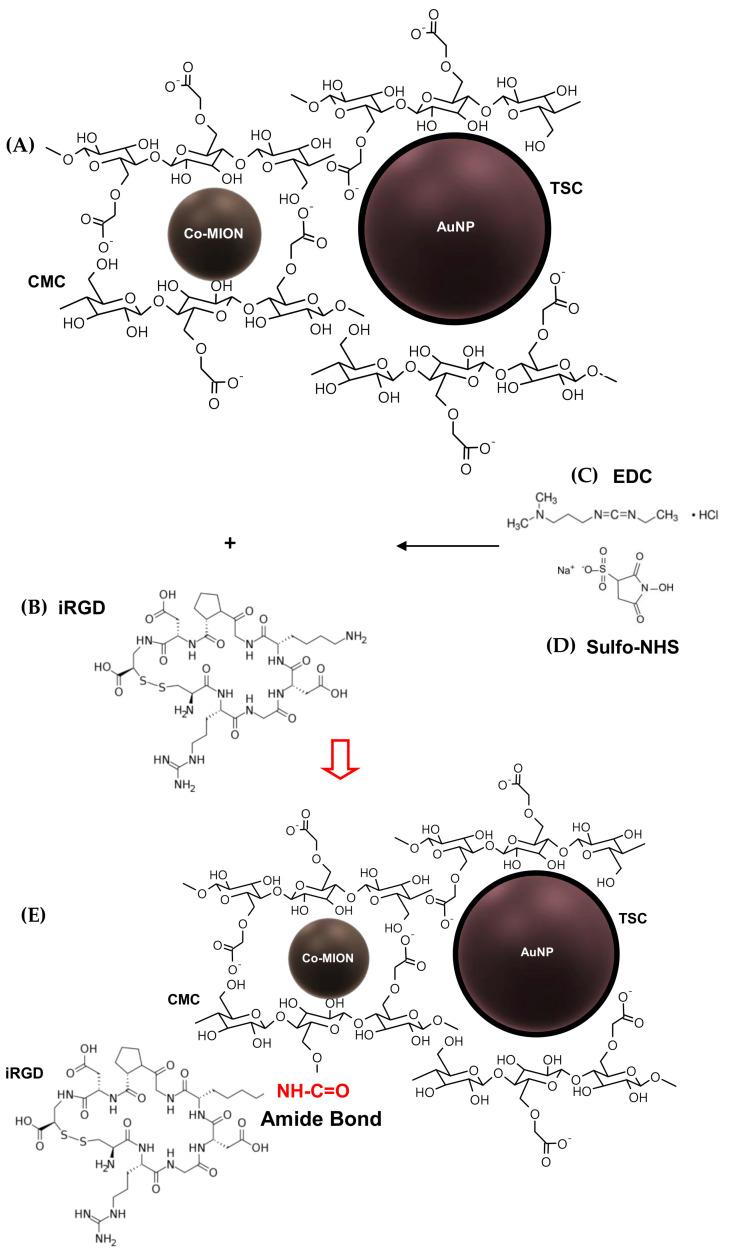
Chemical structure of biopolymer–peptide conjugate precursors: (**A**) inorganic–inorganic dual-nanozyme systems functionalized by a carboxymethylcellulose (CMC) biopolymer organic shell. (**B**) iRGD. (**C**) 1-ethyl-3-(3-dimethylaminopropyl)carbodiimide (EDC). (**D**) N-hydroxysulfosuccinimide sodium salt (sulfo-NHS). (**E**) active integrin-targeting hybrid catalytic nanostructure.

**Figure 2 pharmaceutics-14-02223-f002:**
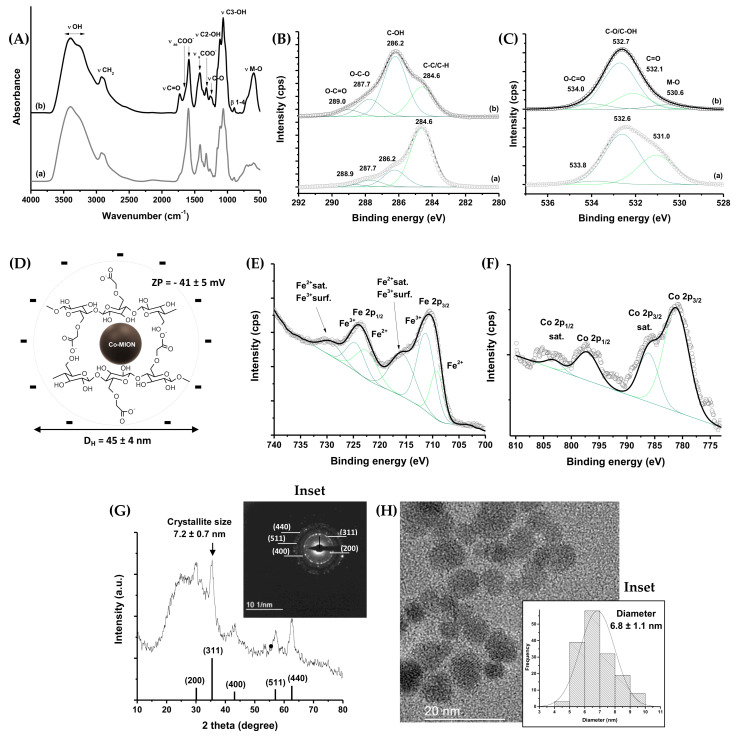
Characterization of Co-MION@CMC: (**A**) FTIR and HR-XPS spectra, and (**B**) C 1s and (**C**) O 1s regions, were obtained from (**a**) CMC polymer and (**b**) Co-MION@CMC nanostructure. (**D**) Schematic representation of the supramolecular structure of the nanosystem, with ZP and D_H_ information. HR-XPS spectra of the (**E**) Fe 2p and (**F**) Co 2p regions. (**G**) XRD pattern with diffraction planes compared to magnetite’s ICCD file (inset: SAED diffraction lines). (**H**) TEM image (inset: histogram of size distribution).

**Figure 3 pharmaceutics-14-02223-f003:**
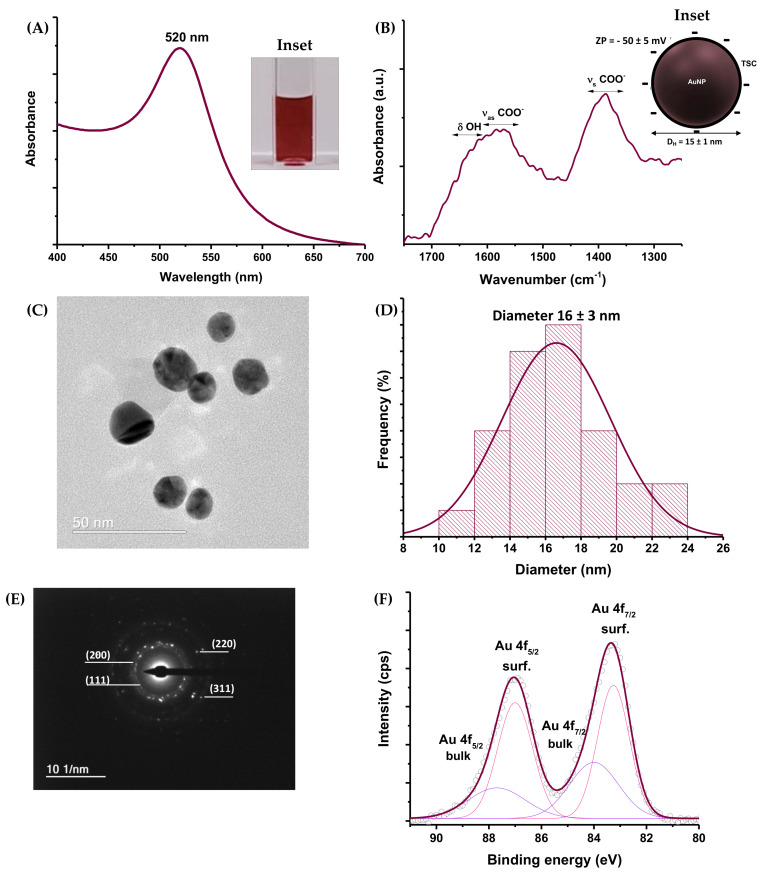
Characterization of AuNPs: (**A**) UV–Vis spectrum (inset: digital image of colloidal suspension). (**B**) FTIR and schematic representation of ZP and D_H_ (inset). (**C**) Nanoparticle image and (**D**) size distribution histogram based on TEM analysis. (**E**) SAED image with diffraction planes. (**F**) HR-XPS spectrum of the Au 4f region.

**Figure 4 pharmaceutics-14-02223-f004:**
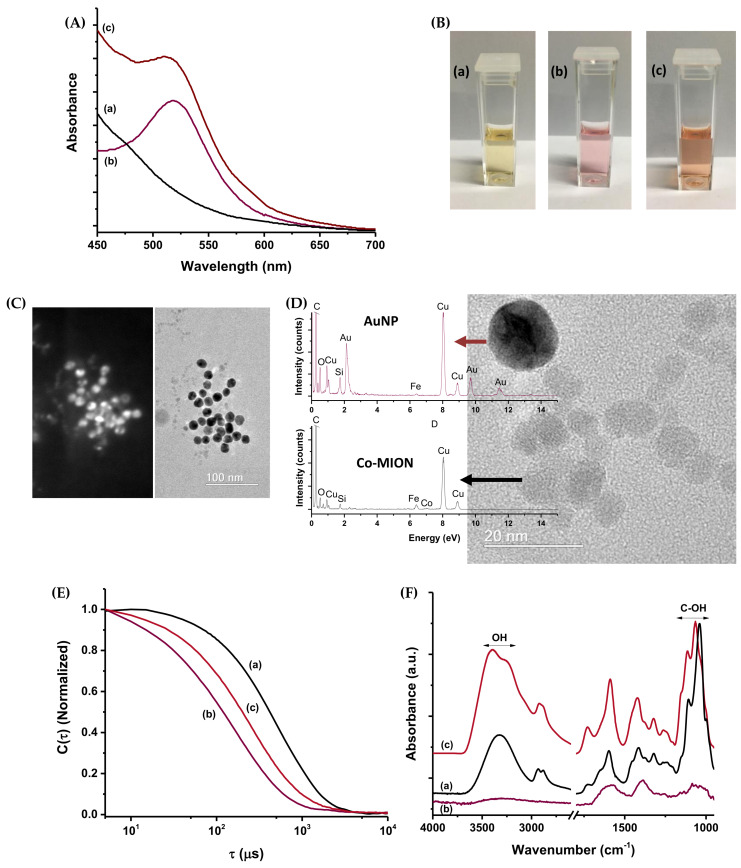
Characterization of the dual-nanozyme system (AuNP//Co-MION@CMC): (**A**) UV–Vis spectra. (**B**) Digital images of suspensions. (**C**) Annular dark-field image (left) and TEM image (right). (**D**) TEM image with EDS spectra. (**E**) Normalized correlation curves. (**F**) FTIR spectra obtained for the dual-nanozyme system (c) compared to Co-MION@CMC (a) and AuNPs (b).

**Figure 5 pharmaceutics-14-02223-f005:**
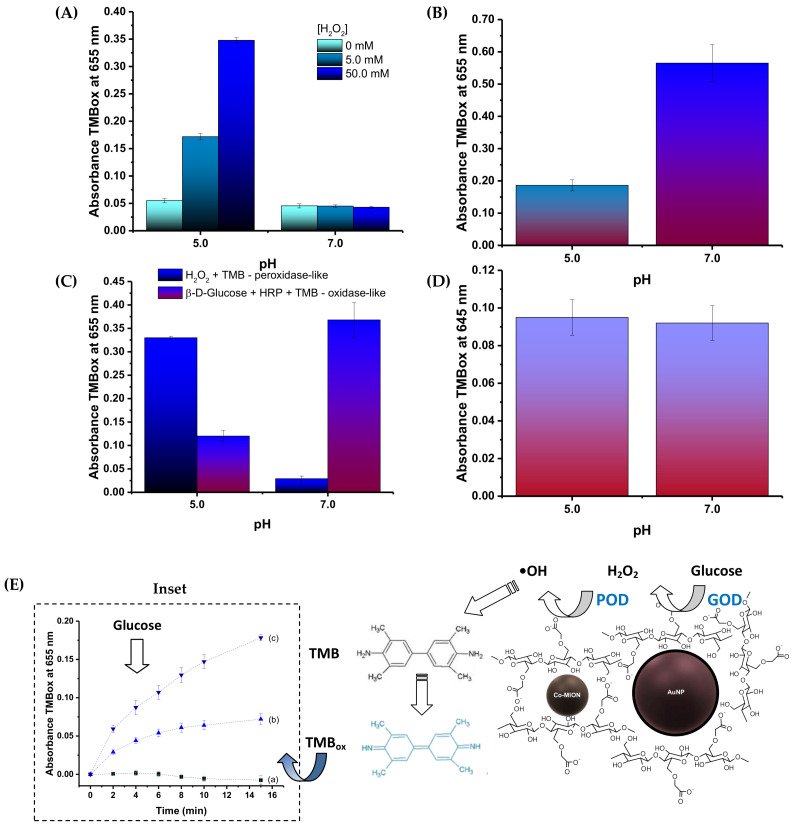
Evaluation of (**A**) peroxidase-like activity of Co-MION@CMC at different concentrations of H_2_O_2_ and (**B**) AuNP oxidase-like behavior (after 60 min). (**C**) GOD and POD activities of AuNPs and Co-MION@CMC after coupling (after 45 min, H_2_O_2_ = 50 mM). (**D**) Catalytic cascade activity based on the two-step inorganic–inorganic dual-nanozyme supramolecular assemblies, leading to TMB oxidation (after 180 min, β-D-glucose = 400 mM; absorbance of control = 0.055 (without dual-nanozyme system)). (**E**) Schematic representation of dual-nanozyme catalytic activity with a plot (inset) of the evolution of absorption of TMB_ox_ over time after incubation with the binanozyme and glucose (the initial substrate of the cascade) at different concentrations (glucose: (**a**) 0 mM, (**b**) 750 mM, and (**c**) 1000 mM; binanozyme: 35 μg/mL//150 μg/mL, AuNP//Co-MION; pH = 7.0).

**Figure 6 pharmaceutics-14-02223-f006:**
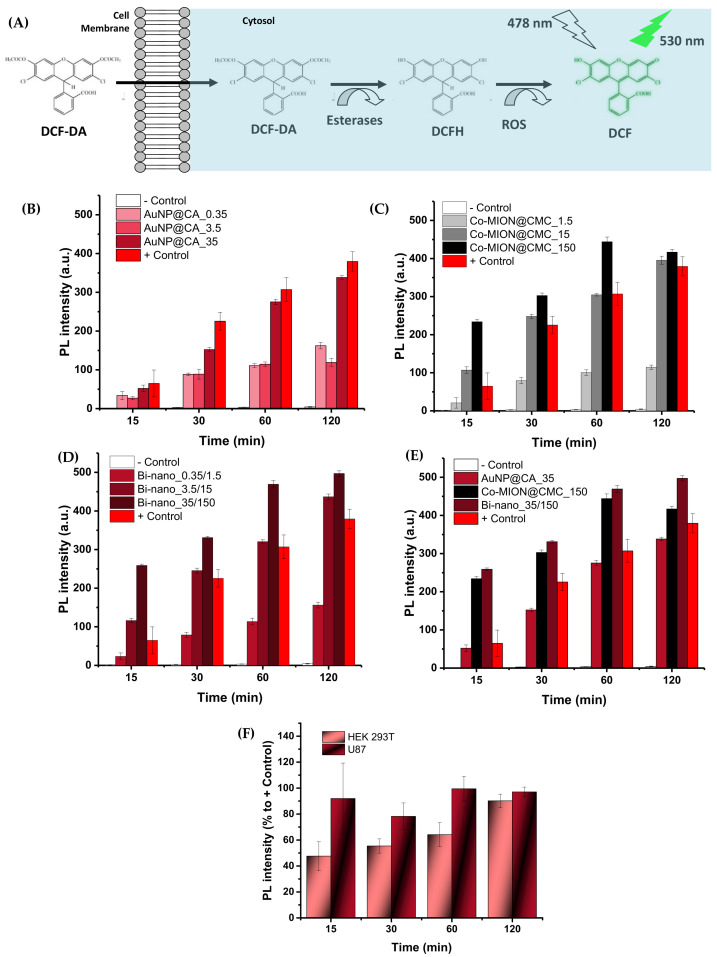
(**A**) Schematic representation of staining of ROS species with DCF-DA. PL intensity associated with the accumulation of intracellular ROS in the U87 MG cell line induced by (**B**) AuNPs, (**C**) Co-MION@CMC, and (**D**) the dual-nanozyme system after 15, 30, 60, and 120 min of exposure and at different concentrations. (**E**) Comparison of the PL intensity of the individual components and the dual-nanozyme system for the highest concentration of each nanozyme as the incubation time increased. (**F**) Effect of cell type measured for the dual-nanozyme system considering % to +Control (binanozyme concentration: 3.5 μg/mL//15 μg/mL, AuNP//Co-MION).

**Figure 7 pharmaceutics-14-02223-f007:**
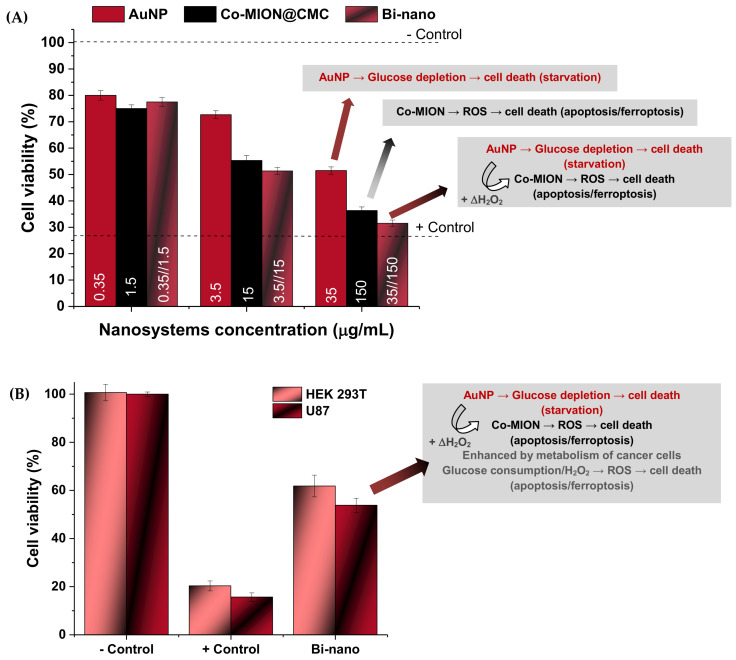
(**A**) Cytotoxicity of nanosystems at different concentrations after incubation for 24 h with the U87 tumor cell line. (**B**) Comparison of cell viability for normal (HEK 239T) and cancer (U87) cells after incubation with the dual-nanozyme catalytic cascade nanoconjugate (Bi-nano concentration: 3.5 μg/mL//15 μg/mL, AuNP//Co-MION).

**Figure 8 pharmaceutics-14-02223-f008:**
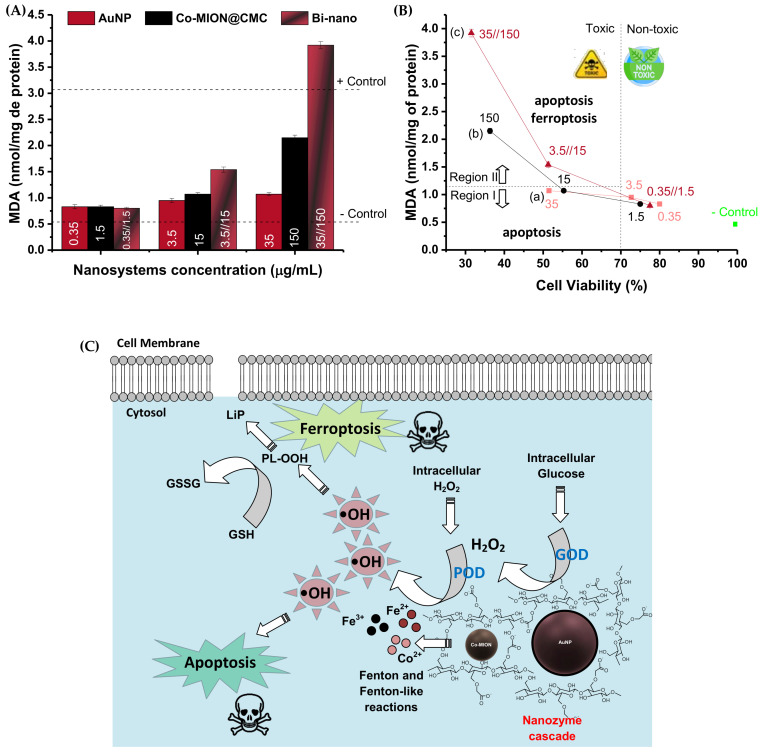
(**A**) Results of lipid peroxidation assessed by malondialdehyde (MDA) levels. (**B**) Comparison of MTT and MDA results for evaluating toxicity mechanisms ((**a**) AuNP, (**b**) Co-MION@CMC, and (**c**) Bi-nano). (**C**) Schematic representation of cell death pathways/mechanisms (e.g., apoptosis and ferroptosis) provoked by nanozyme cascade reactions.

**Figure 9 pharmaceutics-14-02223-f009:**
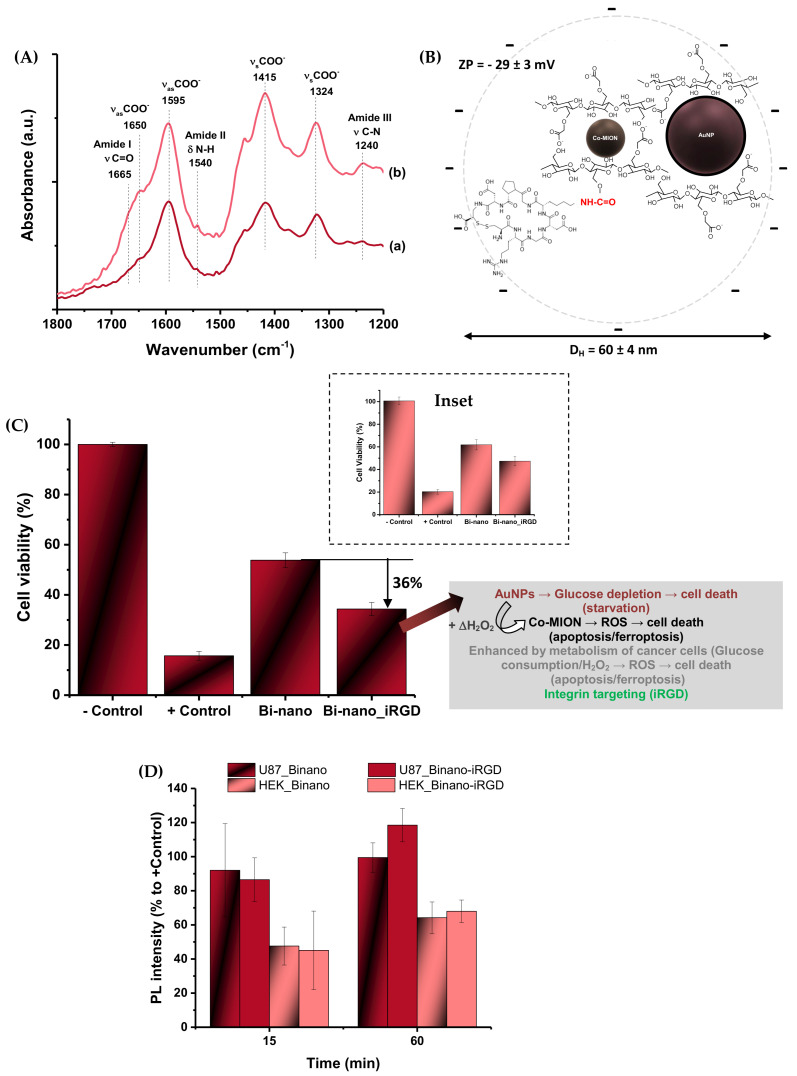
(**A**) FTIR spectra and (**B**) ZP and H_D_ values of Bi-nano upon conjugation with iRGD peptides. (**C**) MTT results of the effects of targeting of the nanostructure by iRGD peptides in contact with the U87 cancer cell line and HEK 293T normal cell line (inset). (**D**) Effects of iRGD conjugation on the intracellular ROS generated by normal and cancer cells after 15 and 60 min of incubation (binanozyme concentration in the MTT and ROS experiments: 3.5 μg/mL//15 μg/mL, AuNP//Co-MION).

**Figure 10 pharmaceutics-14-02223-f010:**
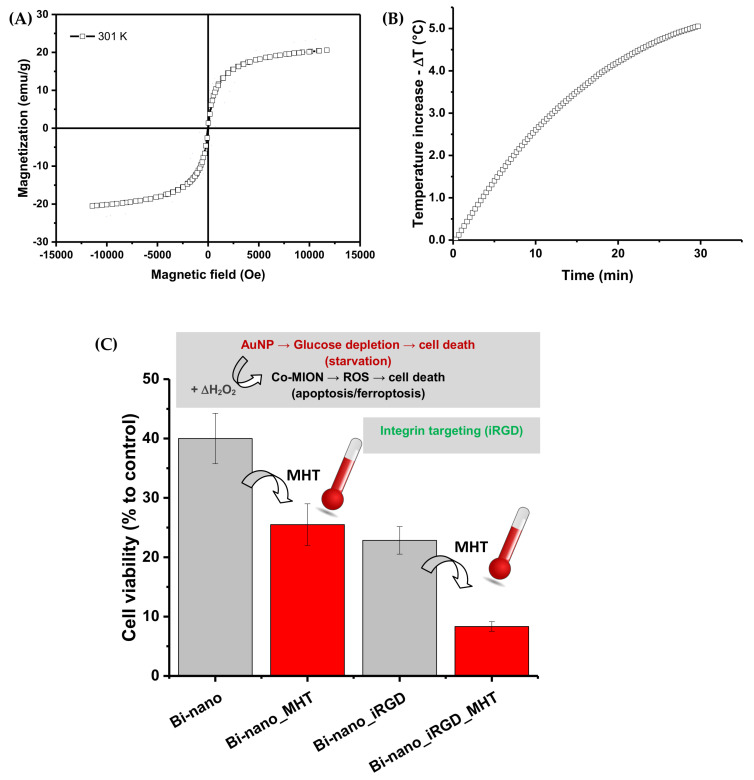
(**A**) Magnetization *versus* magnetic field at room temperature (300 K). (**B**) Heating curve induced by magnetic hyperthermia experiments (∆T = 5.2 °C) associated with Co-MION@CMC nanoparticles. (**C**) Cell viability response of U87 brain cancer cells after incubation with Bi-nano and Bi-nano_iRGD for evaluation of the combined effects of magnetic hyperthermal therapy and chemodynamic biocatalytic therapy (+ Control: 10.3 ± 1.4 %; - Control: 98.8 ± 2.5 %; -Control + MHT: > 90%).

**Figure 11 pharmaceutics-14-02223-f011:**
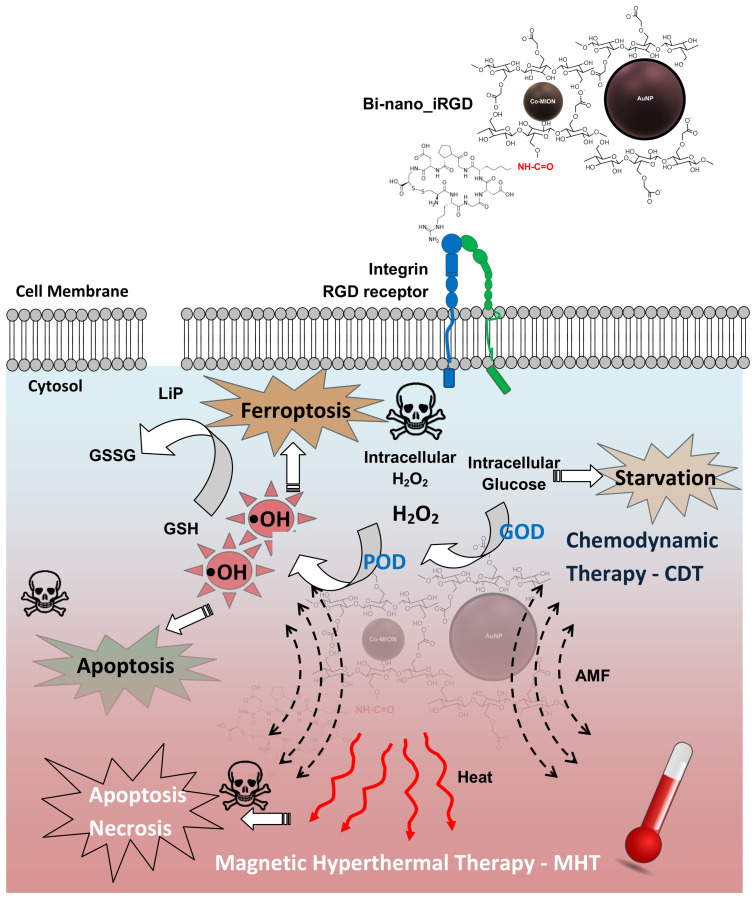
Schematic representation of the developed peptide-functionalized biopolymer magneto-responsive dual-nanozyme hybrid nanoassemblies targeted toward biocatalytic hyperthermal chemodynamic therapy of brain cancer cells.

## Data Availability

All relevant data are available in the manuscript and the Appendix A.

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
