# Peer review of "Bioengineered Carboxymethylcellulose–Peptide Hybrid Nanozyme Cascade for Targeted Intracellular Biocatalytic–Magnetothermal Therapy of Brain Cancer Cells"

_pharmaceutics, 2022, doi:10.3390/pharmaceutics14102223_

Round 1

Reviewer 1 Report

In this paper, a novel cascade enzyme was constructed by coupling gold nanoparticles with cobalt doped superparamagnetic iron oxide nanoparticles. Gold nanoparticles have glucose oxidase characteristics, while cobalt doped paramagnetic iron oxide nanoparticles have Fenton effect and magnetocaloric conversion effect. In addition, the author also grafted iRGD peptide onto the surface of the material to achieve the effect of targeting cancer cells and enhance the lethality to cells. Before this article is published, there are still many problems that need to be noted and modified.

1. The grammar of the article needs to be greatly improved, and the writing logic in the article can be appropriately adjusted.

2. Some information in the pictures in the article is blocked (such as Fig. 3, 7, 10). Please adjust these pictures.

3. In order to make this article more convincing, the article needs to add some necessary references. For example: 1. https://doi.org/10.1186/s12951-022-01278-z. 2. https://doi.org/10.1021/acsnano.7b08868. 3. https://doi.org/10.1021/acsnano.8b06201.

4. In this paper, the authors constructed Co doped superparamagnetic iron oxide nanoparticles. What is the specific purpose of Co doping? What are the advantages over superparamagnetic iron oxide?

5. The nanosystem targeted tumor cells by iRGD peptide, but the specific targeting effect was not shown in this paper. It would be more convincing if the authors could demonstrate the targeting effect of iRGD peptide by laser confocal or streaming methods.

Reviewer 2 Report

In this submission, Mansur et al. report fthe rational design and synthesis of novel hybrid colloidal nanostructures composed of gold nanoparticles stabilized by trisodium citrates (AuNP@TSC), as the oxidase-like nanozyme, coupled with cobalt-doped superparamagnetic iron oxide nanoparticles (Co-MION@CMC), as the peroxidase-like nanozyme. This is an interesting study, but a few mechanisms are not clear in its current format which I have listed below.

- please add basic features of nanosystems like size, surface morphology, surface charge etc in the abstract.

- Please give the examples of carbohydrate-based system in introduction.

- What was the concentration of AUNP 'the Au nanoparticles (termed AuNP) were obtained in an aqueous medium ...'.

- What was the rationale behind choosing the concentrations of nanoformulations? any clinical relevance of such concentrations?

- Authors show cell viability but no information on cellular localisation, internalisation and uptake, any image-based evidences on this? can they quantify how much particles were taken up by cells? cite this paper too; https://doi.org/10.1002/adfm.201903549

- please justify the choice of cell lines?

- could authors please elaborate the potential mechanism of clearance and excretion pathways of such nanoparticles from the living system? they can take help from this paper too; https://doi.org/10.1002/advs.201903441

Reviewer 3 Report

The paper by Alexandra et al. for the first time design and synthesis of novel hybrid colloidal nanostructures as the oxidase-like and peroxidase-like nanozyme, which can trigger biocatalytic cascade reaction in the cancer tumor microenvironments for the combination of magnetothermal-chemodynamic therapy. This work effectively targeting and killing live brain cancer cells in vitro through an enhanced chemocatalytic-magnetothermotherapy process. However, there are several concerns that the authors require to address them.

1. In Fig. 2G and Fig. 3E, The SAED diffraction lines should be labeled, which corresponding to the XRD pattern.

2. In Fig. 2H and Fig. 4C, Element mapping is recommend to further confirm the dispersion of the metal in the nanoparticles.

3. In 3.2.3, The Michaelis-Menten equation of the dual-nanozyme is recommend to reveal the efficiency of catalysis.

4. In 3.3.4, Cancer cell targeting via iRGD should be further verfied with fluorescence imaging.

5. In Fig. 10B, Please provide the thermal imaging of Co-MION@CMC nanoparticles. In addition, what about the stability of Co-MION@CMC nanoparticles.

6. Some grammar and spell shoule be checked especially in the experimental section.

Round 2

Reviewer 1 Report

The author answered the questions, and I think this article is acceptable and published in this journal.

Reviewer 2 Report

Authors have elegantly addressed all the concerns raised during the first round of review. I am pleased to recommend the revised manuscript for publication in Pharmaceutics.

Just one comment, reference 57 has two references under one number, could please authors make them two independent references.